# Metabolic activity and survival strategies of thermophilic microbiomes during hyperthermophilic composting

Chen Liu,[1] Yuqi He,[1] Hongbo Zhang,[1] Dong Zhang,[1] Chaofan Ai,[1] Xiang Tang,[1] Qiu-e Yang,[1] Zhen Yu,[2] Shiyong Tan,[3] Ville-Petri Friman,[4] Hanpeng Liao,[1] Shungui Zhou[1]

**ABSTRACT** Hyperthermophilic composting (HTC) is a promising strategy for the treatment of organic solid waste, leveraging extreme thermophilic conditions (up to 90°C) driven by specialized microbial communities. While microbial community composition and succession have been previously described during HTC, the metabolic activity and adaptation of thermophilic microbiomes remain largely unexplored. In this study, we conducted time-series metagenomic and metatranscriptomic analyses on samples from a full-scale HTC system to characterize the composition, functional potential, and metabolic activity of thermophilic bacteria. A total of 227 non-redundant metagenome-assembled genomes (MAGs) were recovered, including 45 thermophilic MAGs (optimal growth temperatures > 45°C). Metatranscriptomic profiling revealed that thermophilic taxa—such as *Thermus thermophilus*, *Planifilum fulgidum*, and *Thermaerobacter* spp.—were highly transcriptionally active and played vital roles in heat generation through the upregulation of energy production and carbohydrate metabolism pathways. Additionally, these thermophiles exhibited survival and adaptation strategies involving physiological changes (e.g., spore formation, enhanced motility, and genome streamlining) and the induction of thermal resistance mechanisms (e.g., DNA repair systems, heat-shock proteins, and synthesis of compatible solutes). Overall, this study provides novel insights into the diverse survival strategies of thermophilic microbiomes in HTC and suggests potential avenues for optimizing thermophilic biotreatment processes for solid waste management.

**IMPORTANCE** Despite increasing interest in hyperthermophilic composting as a sustainable waste treatment strategy, the mechanisms by which microbial communities both tolerate and drive extreme thermal conditions remain unclear. This study fills a critical knowledge gap by identifying a small group of highly active thermophilic bacteria that dominate during peak composting temperatures and orchestrate endogenous heat production. Using genome-resolved multi-omics, we demonstrate that these thermophiles couple high metabolic output with specialized survival strategies—such as genome streamlining, thermotolerance systems, and adaptive motility systems. These findings advance our understanding of microbial function under extreme conditions and provide a framework for optimizing thermophilic microbiome performance in engineered ecosystems.

**KEYWORDS** metatranscriptomics, metagenomics, hyperthermophilic composting, microbiome, metabolic activities

Composting is an effective method for treating solid waste, converting various materials—such as food, municipal, and agricultural waste—into organic fertilizer (1). This microbially driven biochemical process decomposes organic matter, generating heat, with fermentation temperatures typically exceeding 60°C (1, 2). The composting

**Peer Reviewer** Linwei Wu, Peking University, Beijing, China

Address correspondence to Hanpeng Liao, liaohp@fafu.edu.cn.

Chen Liu and Yuqi He contributed equally to this article. The author order was determined by drawing straws.

The authors declare no conflict of interest.

See the funding table on p. 14.

process is generally divided into three phases based on temperature: the initiation phase (ambient temperature), the thermophilic phase (high temperature), and the maturation phase (return to ambient temperature). The composition and diversity of microbial communities across these stages have been extensively studied (3–5). Notably, microbial community structure shifts significantly with temperature fluctuations (6, 7). As temperatures rise, heat-resistant bacteria (e.g., Firmicutes) rapidly dominate (5), while during the maturation phase, the decline in readily degradable substrates allows heat-sensitive taxa such as Actinobacteria to predominate (5). Although microbial community dynamics during composting have been well characterized using amplicon sequencing (3–5), the composition and roles of thermophilic microbiomes at the community level remain unclear.

Temperature is a key driver of both organic matter decomposition and microbial community succession during composting. High temperatures not only eliminate pathogens in manure but also accelerate humification processes (8). Traditionally, thermophilic composting operates optimally at 60°C–70°C. However, in 2008, hyperthermophilic composting (HTC) technology was developed, achieving temperatures above 90°C without external heating, which is 20°C–30°C higher than conventional composting (9). Under these extreme conditions, the bacterial community in HTC is often enriched with extreme thermophiles such as *Thermus* and *Planifilum*, a community feature that coincides with elevated temperatures (10, 11). HTC can maintain high microbial activity under extreme thermal stress, thereby enhancing organic matter degradation and humus formation (8, 12). While the diversity and composition of thermophilic bacterial and viral communities during the HTC have been studied (7, 8, 11), the metabolic activity of these thermophilic microorganisms during the HTC process, as well as their role in heat production during composting, remains poorly understood.

Heat production in composting arises from the accumulation of metabolic heat from microbial activity, while in turn, temperature shapes microbial composition and function. Although survival strategies of thermophilic bacteria have been extensively investigated in laboratory settings using cultured isolates, their roles and adaptations in complex environments like HTC are less understood (13, 14). Recent advances in genome-resolved metagenomics and metatranscriptomics have enabled researchers to reconstruct microbial genomes and characterize transcriptional activity in extreme environments such as hot springs and hydrothermal vents (15–18). These studies have uncovered a wide array of survival strategies employed by thermophilic microorganisms, including sporulation, stress response activation, and shifts in cellular behavior, which reflect long-term evolutionary adaptation to persistent thermal stress (17, 19, 20). In contrast, HTC is an artificial extreme environment formed over a relatively short period (~30 days), and it remains unclear how microbial communities rapidly adapt to such conditions. Specifically, the metabolic activity and heat tolerance mechanisms of thermophilic microorganisms in HTC remain poorly understood.

To address these gaps, we employed time-series metagenomic and metatranscriptomic sequencing to investigate the metabolic and survival strategies of thermophilic bacterial communities in full-scale HTC plants. Our objectives were to (i) identify the key thermophilic taxa and metabolic pathways responsible for heat production, and (ii) characterize the functional activity and adaptive strategies of thermophiles throughout the HTC process. We hypothesized that (i) core thermophilic taxa act as primary drivers of heat production and (ii) thermophilic bacteria employ a range of physiological and molecular strategies to cope with extreme thermal stress during HTC, such as sporulation, shifts in cellular behavior, and activation of stress response pathways. This study provides genome-resolved insights into the metabolic heat generation and survival strategies of key thermophiles in HTC, with implications for optimizing microbial processes in high-temperature organic waste treatment systems.

## MATERIALS AND METHODS

### HTC experiments

In this study, full-scale HTC experiments were conducted using composting material at GeoGreen Innotech Co., Ltd., in Beijing, China, as previously described (6). The raw materials were also sourced from the same facility, which consisted of dehydrated sewage sludge (80% water content [WC]), rice husk (13% WC), and composting by-products (40% WC) under the 1:1:4 ratio (by weight). These materials were thoroughly mixed using a forklift, resulting in an overall moisture content of 60%. The mixed composting material (approximately 200 tons) was loaded into independent compost compartments with 8.5 m in length, 6 m in width, and 3.5 m in height. To ensure adequate aeration, two aeration pipes were installed at the bottom of each compartment, each delivering air at a rate of 2 $m^3$/h. Composting temperatures were automatically monitored (PT100; Shanghai Chekon Instrument Co., Ltd., China), with the temperature reaching 90°C within 24 hours after the start of fermentation. The HTC process was carried out over 45 days, during which four distinct composting phases were identified based on temperature dynamics: the initial phase (days 0–1 [D0–1]), the hyperthermophilic phase (D2–14), thermophilic phase (D15–26), and maturation phase (D27–45). The experiment was performed during the summer of 2020 with five independent replicates.

### Sample collection and analysis of compost physicochemical properties

The HTC experiment lasted for 45 days and samples were taken at days 0, 4, 9, 15, 21, 27, 33, and 45 after the initiation of the HTC experiment, representing different stages of the composting process. To maintain consistency, all samples were taken from the top 50 cm of the compost pile. For each composting pile, five subsamples were collected—one from the center and four from each corner—and thoroughly mixed to form a composite sample. Each composite sample was divided into two parts: one stored at 4°C for physicochemical analyses and the other stored in liquid nitrogen for microbial community analysis. The physicochemical properties of the compost, including WC, ammonium ($NH_4^+$), nitrate ($NO_3^-$), total organic carbon (TOC), dissolved total nitrogen (DN), inorganic carbon (IC), total carbon (TC), total nitrogen (TN), electrical conductivity (EC), and pH, were measured following previously described methods (21). WC was determined by drying compost samples at 105°C for 24 hours using the drying oven. $NH_4^+$ and $NO_3^-$ were extracted using 2 M KCl and measured with a continuous-flow autoanalyzer after a 10-fold dilution (FlowSys, Systea, Rome, Italy). TOC, DN, and IC were extracted using deionized water at a 1:10 (vol:vol) ratio and analyzed with an automated TOC analyzer (Shimadzu TOC-L CPH, Kyoto, Japan). TC and TN were measured via dry combustion using an Elementar CHNS Analyzer (Vario MAX cube, Hanau, Germany), and the C/N ratio was calculated accordingly. EC and pH were measured using a pH meter with deionized water at a 1:10 (vol:vol) ratio for the extraction. All samples were stored at 4°C for no longer than 2 weeks and were processed for physicochemical analyses as soon as possible after collection.

### 16S rRNA gene high-throughput sequencing and analysis

Compost samples collected on days 0, 4, 9, 15, 21, 27, 33, and 45 (with five replicates per time point) were used for 16S rRNA gene high-throughput sequencing. Total microbial DNA was extracted from 0.25 g of compost samples using the Soil DNA Extraction Kit (mCHIP, Guangzhou, China) according to the manufacturer's instructions. DNA concentration and quality were assessed using the Qubit dsDNA High Sensitivity Assay Kit. After electrophoresis and concentration analysis, the qualified DNA samples were sent to Guangdong Magigene Biotechnology Co., Ltd. (Guangzhou, China) for high-throughput 16S rRNA sequencing using primers 515F (5′-GTGCCAGCMGCCGCGGTAA-3′) and 907R (5′-CCGTCAATTCMTTTRAGTTT-3′). These primers amplify the V4–V5 region of the

16S rRNA gene, which provides broad coverage and reliable taxonomic resolution for bacterial community profiling (22). More details on data analysis are provided in Text S1.

## Metagenomic sequencing and assembly analysis

To investigate microbial functional changes during HTC, three randomly selected replicate samples ($n$ = 3) were collected at days 0, 4, 15, and 27, corresponding to the initial, hyperthermophilic, thermophilic, and maturation phases, respectively. Total microbial DNA was extracted using the same kit as in 16S rRNA high-throughput sequencing and sent for shotgun metagenomic sequencing on the Illumina HiSeq X Ten platform (PE150 mode; Guangdong Magigene Biotechnology Co., Ltd.). Shotgun metagenomic sequencing generated approximately 60–120 million raw reads per sample, corresponding to 7.3–17.7 Gbp per library (Table S1). Genomic DNA was randomly fragmented into ~300 bp segments, and sequencing libraries were constructed using ~1 µg of DNA with the NEBNext DNA Library Prep Kit (Illumina), following the manufacturer's instructions. Metagenomic read processing and analysis were performed as previously described (11). Briefly, read quality was assessed with FastQC v0.11.2 and low-quality reads (Phred score < 30 or read length < 36 bp) were removed using Trimmomatic v0.39 (23). Clean reads from each composting phase were assembled independently using SPAdes v3.13.1 with the "--meta" flag and k-mer sizes 33, 55, 77, 99, 111, and 121 (24).

Contigs from all metagenomes were clustered with CD-HIT (85% coverage, 90% identity), and the resulting non-redundant contigs were subjected to gene prediction and functional annotation using Diamond v0.97 against the NCBI-nr and Kyoto Encyclopedia of Genes and Genomes (KEGG) databases. Microbial genome reconstruction was performed using the metaWRAP pipeline v1.1.1 (25). Contigs >2 kb were binned using MetaWRAP's binning module, and raw bins were refined to improve genome quality using the Bin_refinement module. Dereplication of all refined bins across samples was done with dRep v2.3.2 (26). Only bins with ≥50% completeness and <10% contamination, as assessed by CheckM v1.0.1 (27), were retained for downstream analyses. Taxonomic classification of bins was performed using GTDB-Tk (v2.3) of classify_wf pipeline, based on 120 conserved bacterial single-copy marker genes and the Genome Taxonomy Database (release 03-RS86) (28). To investigate the functional potential of thermotolerant microbes, we predicted the optimal growth temperatures (OGTs) of the reconstructed metagenome-assembled genomes (MAGs) using the machine learning-based tool Tome v1.1 (29). MAGs with OGTs ≥45°C were classified as thermophilic, while those with OGTs <45°C were classified as mesophilic. Open reading frames in the assembled contigs and MAGs were predicted using Prodigal v2.6 with the "-p meta" option, and functionally annotated against the NCBI-nr, KEGG, and dbCAN databases using Diamond v0.97 (e-value < $1^{e-5}$) (30, 31).

To assess changes in the abundance of MAGs and functional genes in compost metagenomes, quality-filtered (QC-passed) reads were mapped to contigs or gene sequences to generate abundance profiles, following the method described by Liao et al. (11). Briefly, metagenomic reads from the four composting phases (each with three biological replicates, totaling 12 samples) were mapped to MAG sequences and target genes using the CoverM pipeline. CoverM was executed in "genome" mode for MAGs and "contig" mode for genes, with abundance calculated using the transcripts per kilobase per million mapped reads (TPM) method. TPM values were used for inter-sample comparisons, as they account for both gene length and sequencing depth. Relative abundances of all MAGs across samples were compiled into abundance matrices, which were subsequently used for downstream analyses, including alpha and beta diversity assessments.

## RNA extraction and metatranscriptomics analysis

To investigate the transcriptional activity of MAGs and their functional genes during HTC, RNA shotgun sequencing was performed on compost samples collected at days

0, 4, 15, and 27 ($n$ = 3 per phase). Total RNA was extracted from 2 g of compost using the RNAsimple Total RNA Extraction Kit (Tiangen, Beijing, China) following the manufacturer's instructions. Ribosomal RNA (rRNA) was removed using the RiboMinus Transcriptome Isolation Kit (Thermo Fisher Scientific), and residual DNA was eliminated through DNase treatment (Vazyme, Nanjing, China) until no DNA contamination was detected by PCR using 16S rRNA gene primers. Library construction was performed using the TruSeq Stranded mRNA Library Prep Kit (Illumina, California, USA) according to the manufacturer's protocol. All the above experimental processes, including RNA extraction, library construction, sequencing, etc., were carried out by Guangdong Magigene Biotechnology Co., Ltd. (Guangdong, China). Shotgun sequencing was performed on the Illumina NovaSeq 6000 platform (PE150 mode). In total, 12 compost samples generated a total of 120 Gb of metatranscriptomic data set (10 Gb per sample).

Quality assessment and control of raw reads were performed using FastQC v0.11.2 with default parameters and Trimmomatic v0.39 (Phred score >30 and read length >50 bases), respectively. Although non-coding rRNA was experimentally removed prior to library construction, residual rRNA-associated reads were further removed using SortMeRNA v4.3.4 (23, 32). The remaining mRNA reads were used to estimate the transcriptional activity of MAGs and their functional genes by mapping them to MAG contigs and gene sequences using the CoverM pipeline in "genome" mode, with TPM as the normalization method. Likewise, the transcriptional activities of functional genes within MAGs were assessed based on metatranscriptomic coverage by mapping the filtered mRNA reads to gene sequences using CoverM in "contig" mode. Differential gene expression analysis between composting phases was conducted using DESeq2, with a false discovery rate-adjusted $P$-value threshold of <0.05 (33).

## Statistical analyses

All statistical analyses and most visualizations were conducted in the R environment using the RStudio platform (v4.2.2, https://www.r-project.org/). One-way analysis of variance (ANOVA) was performed to test for significant differences when the data met the assumption of normality. Differences in microbial community composition across composting stages were assessed using non-parametric PERMANOVA (adonis function, 999 permutations) based on a distance matrix. Principal Coordinates Analysis (PCoA) based on the weighted UniFrac distance and alpha-diversity (Richness and Shannon index) based on ASVs table were performed using the vegan and ggvegan package in R.

## RESULTS AND DISCUSSION

### The succession of the bacterial community is related to heat production during HTC

Changes in composting temperature were monitored during a 45-day full-scale HTC experiment. The temperature increased rapidly, reaching a peak of 94°C within 3 days and remained above 80°C for 9 days during the hyperthermophilic phase (D4). It then gradually declined to 55°C during the thermophilic phase (D15) and approached ambient levels by day 27, marking the maturation phase (D27; Fig. S1). To investigate bacterial community dynamics across the different phases of HTC, 16S rRNA gene amplicon sequencing was performed. Consistent with previous findings (6), both bacterial community composition (based on UniFrac distances, PERMANOVA $P$ = 0.001) and diversity ($P$ < 0.01) were closely associated with composting temperature shifts (Fig. 1a and b). At the initial phase (D0), Proteobacteria and Bacteroidetes were dominant, accounting for 62.4% of the community. As temperatures increased, thermophilic phyla such as Firmicutes and Deinococcus-Thermus became more prevalent, rising from 8.4% at D0 to 95.7% by D15. At the genus level, thermophiles such as *Thermus* and *Planifilum* emerged as dominant taxa during the hyperthermophilic phase, with their relative abundances increasing 36-fold at D4 and 169-fold at D15 compared to D0 (Fig. 1c). These genera, known for their strong heat tolerance (6, 11), likely contribute

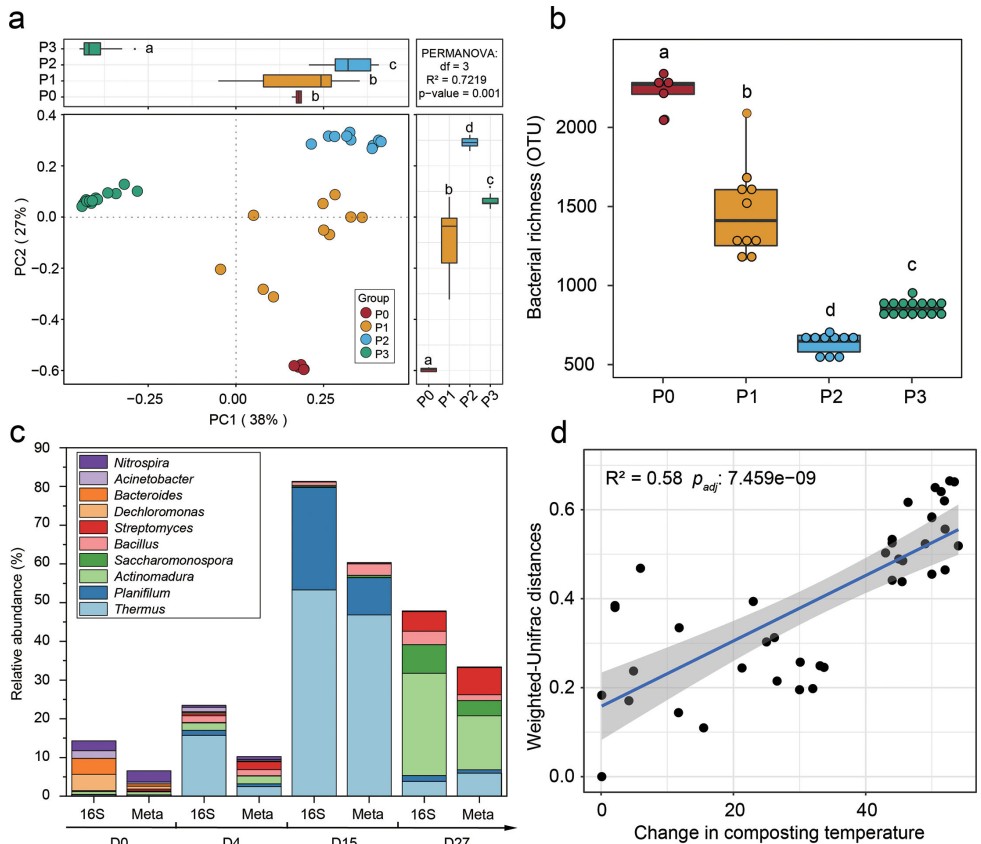

**FIG 1** Dynamics of bacterial community composition and diversity and their relationship with composting temperature during HTC. (a) PCoA based on weighted UniFrac distances shows shifts in bacterial community composition across different HTC phases (Adonis test, $P = 0.001$; based on amplicon sequencing). (b) Changes in bacterial community richness across composting phases based on amplicon sequencing. (c) Relative abundance of the top 10 dominant bacterial genera at different composting phases measured by 16S rRNA gene amplicon sequencing (*x*-axis labeled "16S") and shotgun metagenomics (*x*-axis labeled "meta"). (d) Correlation between bacterial community dissimilarity (weighted UniFrac distance to Day 0) and compost temperature over time during HTC. In panels a and b, P0, P1, P2, and P3 represent the initial, hyperthermophilic, thermophilic, and maturation phases, respectively. The initial phase includes samples from day 0; the hyperthermophilic phase includes samples from days 4 and 9; the thermophilic phase includes samples from days 15 and 21; and the maturation phase includes samples from days 27 and 45. Each time point comprises five biological replicates. In panel c, D0, D4, D15, and D27 refer to samples collected on days 0, 4, 15, and 27 of the composting process.

significantly to the temperature increase during HTC. Microbial community dissimilarities (weighted UniFrac) were strongly correlated with changes in composting temperature ($P < 0.001$), indicating that bacterial community succession is closely associated with heat production during HTC (Fig. 1d). Mantel tests based on Bray–Curtis dissimilarities further showed that other physicochemical parameters—such as nutrient content, pH, and moisture—also exhibited significant associations with bacterial community composition (Fig. S2). These results indicate that multiple factors influence microbial succession during HTC. Nevertheless, temperature remains the most direct determinant of microbial survival and metabolic activity in composting systems, especially during the hyperthermophilic phase (34). Temperature is a critical factor in determining the hygienic safety and decomposition efficiency of organic waste during composting (35). While previous studies have achieved ultra-high composting temperatures via external heating (36, 37), such approaches are energy-intensive and economically costly. Our HTC used thermophiles to achieve and sustain extreme thermophilic conditions, thereby enhancing organic matter degradation and pathogen inactivation compared to traditional composting (34, 38). Taken together, these results indicate that the bacterial

community structure undergoes dramatic shifts throughout the stages of HTC composting, with thermophilic bacteria playing a crucial role in producing extreme high temperatures.

## Bacterial activity is associated with composting temperature increase during HTC

To link composting temperature dynamics with microbial functional changes during HTC, we applied a multi-omics approach combining metagenomics and metatranscriptomics to uncover microbial functional shifts. Genes were annotated from all co-assembled metagenomic contigs >1,000 bp and classified into functional categories based on the KEGG database. A total of 7,427 KEGG orthologs (KOs) were identified across all samples, encompassing 40 level-2 KEGG pathways (Table S2). PCoA revealed that the composition of microbial functions based on KO profiles was significantly influenced by the composting phase, driven by temperature (ANOSIM, $P < 0.01$; Fig. S3). During the thermophilic phase, functions associated with environmental information processing, genetic information processing, cellular processes, and metabolism were significantly enriched compared to the initial phase (D0; Fig. S4), suggesting that high temperatures enhance the microbial community's metabolic capacity. Interestingly, pathways related to human diseases showed a significant reduction in abundance during the thermophilic phase compared to the initial phase (Fig. S3), likely due to the efficient inactivation of pathogens and antibiotic resistance genes at temperatures exceeding 80°C (6). Specifically, 11 KEGG pathways were significantly upregulated during the thermophilic phases (D4 and D15), including those involved in amino acid metabolism, energy metabolism, nucleotide metabolism, membrane transport, metabolism of cofactors and vitamins, lipid metabolism, glycan biosynthesis, translation, and replication and repair (Fig. S5). These findings are consistent with previous observations in garden waste composting, where microbial functions related to cell growth and proliferation were enriched relative to raw compost material (39). As thermophiles typically exhibit faster growth rates than mesophiles, they demand more efficient systems for energy and nutrient metabolism (40). The functional gene enrichment was primarily contributed by Firmicutes, Deinococcus-Thermus, and Proteobacteria (Fig. S6).

To evaluate bacterial metabolic activity during HTC, metatranscriptomic mRNA reads from different composting phases were mapped to the metagenomic contigs. Approximately 81% of quality-filtered mRNA reads successfully mapped back to the assembled contigs, indicating high RNA quality and accurate representation of active microbial communities. Among all transcripts, 55.0% were derived from bacteria, 0.15% from archaea, 0.32% from fungi, 0.21% from viruses, and 44% remained unclassified at the kingdom level (Fig. S7). These results confirm that bacteria are the principal active microbial group during HTC, in agreement with previous findings (39). Taxonomic assignment of transcripts at the phylum and genus levels revealed that Actinobacteria, Firmicutes, Deinococcus-Thermus, Proteobacteria, and Chloroflexi were the dominant active phyla throughout HTC, collectively accounting for ~95% of total transcripts (Fig. 2a). Firmicutes and Deinococcus-Thermus showed elevated transcriptional activity during the thermophilic phase compared to other phases. At the genus level, *Bacillus* and *Bhargavaea* were dominant at D4, while *Thermus*, *Planifilum*, *Thermoactinomyces*, and *Thermaerobacter* exhibited the highest transcriptional activity at D15 (Fig. 2b). The composition of active microbiomes inferred from metatranscriptomics was consistent with taxonomic profiles based on 16S rRNA amplicon sequencing during HTC. Notably, nearly half of all transcripts could not be taxonomically assigned, indicating the presence of many yet-unknown active microorganisms in HTC. The proportion of these unclassified transcripts increased significantly during the thermophilic phase, suggesting that extreme temperatures promote the emergence and activity of uncharacterized heat-tolerant microbes. Taken together, the observed shifts in bacterial metabolic activity across the composting phases closely reflect the temperature dynamics throughout

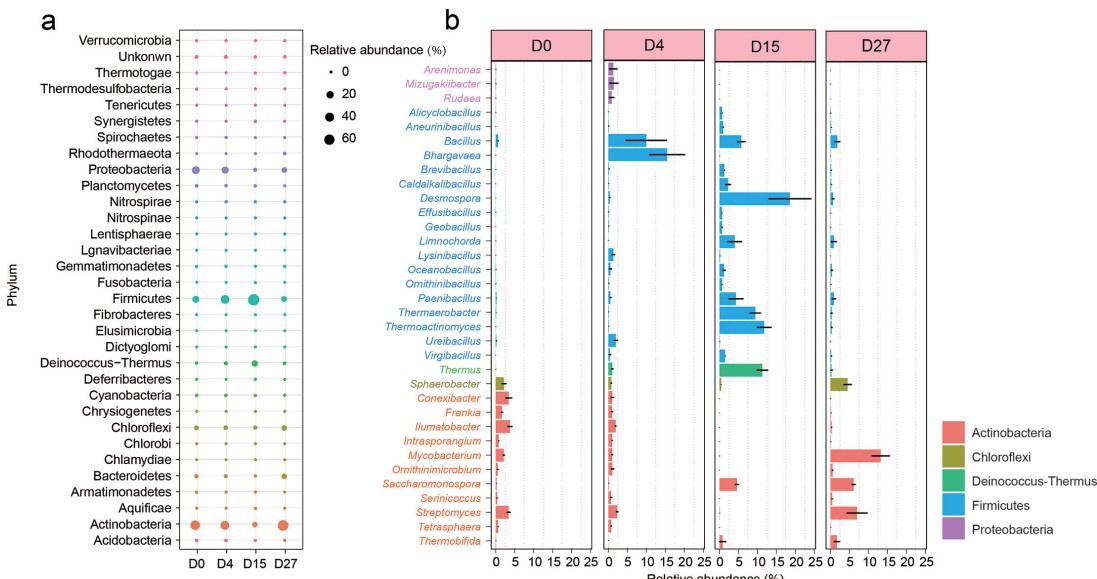

**FIG 2** Temporal changes in the metabolic activity of the bacterial community during HTC at the phylum (a) and genus (b) levels, based on the mean transcriptional activity of functional genes derived from assembled contigs. Each contig was taxonomically classified and used to determine transcriptional activity at different taxonomic levels. (a) Heatmap showing the relative transcriptional activity of functional genes in contigs assigned to bacterial phyla across different composting stages. (b) Boxplots showing the relative transcriptional activity of functional genes in contigs assigned to bacterial genera. The color of the *y*-axis text indicates the corresponding phylum classification for each genus. Gene transcriptional activity was calculated as transcript abundance (transcripts per million, TPM), normalized by gene length and total number of mapped reads. Sampling time points include D0 (initial phase), D4 (hyperthermophilic phase), D15 (thermophilic phase), and D27 (maturation phase).

the HTC process. Notably, a small group of key thermophilic bacteria dominates heat production and metabolic activity during the high-temperature phase.

## Key thermophilic MAGs play a significant role in heat production during HTC

To better understand bacterial metabolic functions at the community level during HTC, we reconstructed MAGs from the assembled metagenomic contigs. In total, we recovered 517 medium- to high-quality MAGs across all composting metagenomes, including two archaeal and 515 bacterial MAGs (completeness ≥ 50% and contamination ≤ 10%). After de-replication, 227 non-redundant bacterial MAGs were retained (Table S3), primarily belonging to the phyla Proteobacteria (55 MAGs), Firmicutes (54), Actinobacteriota (48), Bacteroidota (25), and Chloroflexota (9). The taxonomic composition of MAGs was largely consistent with 16S rRNA gene amplicon sequencing results, indicating that the reconstructed MAGs accurately represent the bacterial diversity during composting. It is worth noting that 195 MAGs could not be classified to any known genus using the current GTDB-Tk database, indicating a substantial number of unknown bacterial groups in HTC. Compared with traditional isolation and culture methods, culture-independent metagenomics provides genomic insights into a vast diversity of uncultivated microbiomes.

To evaluate the thermal adaptation of recovered MAGs, we predicted their OGTs using a machine learning model (29). The predicted OGTs ranged from 24°C to 76°C, comprising 182 mesophilic MAGs (OGT < 45°C) and 45 thermophilic MAGs (OGT > 45°C), including five classified as extremely thermophilic (OGT > 70°C; Table S3). Notably, 82% of the thermophilic MAGs belonged to the Firmicutes phylum (Fig. 3), reinforcing the role of Firmicutes as dominant thermophiles during HTC. For instance, two *Calditerricola* MAGs (D4-3-bin.10, OGT 76.3°C; D4-1-bin.22, OGT 74.7°C) and one *Limnochordales* MAG (D15-1-bin.18, OGT 70.1°C), all affiliated with *Firmicutes*, were among the most heat-tolerant. This aligns with previous studies reporting the widespread occurrence of extreme thermophiles in *Firmicutes*, which are known for their robust stress resistance (41, 42). In

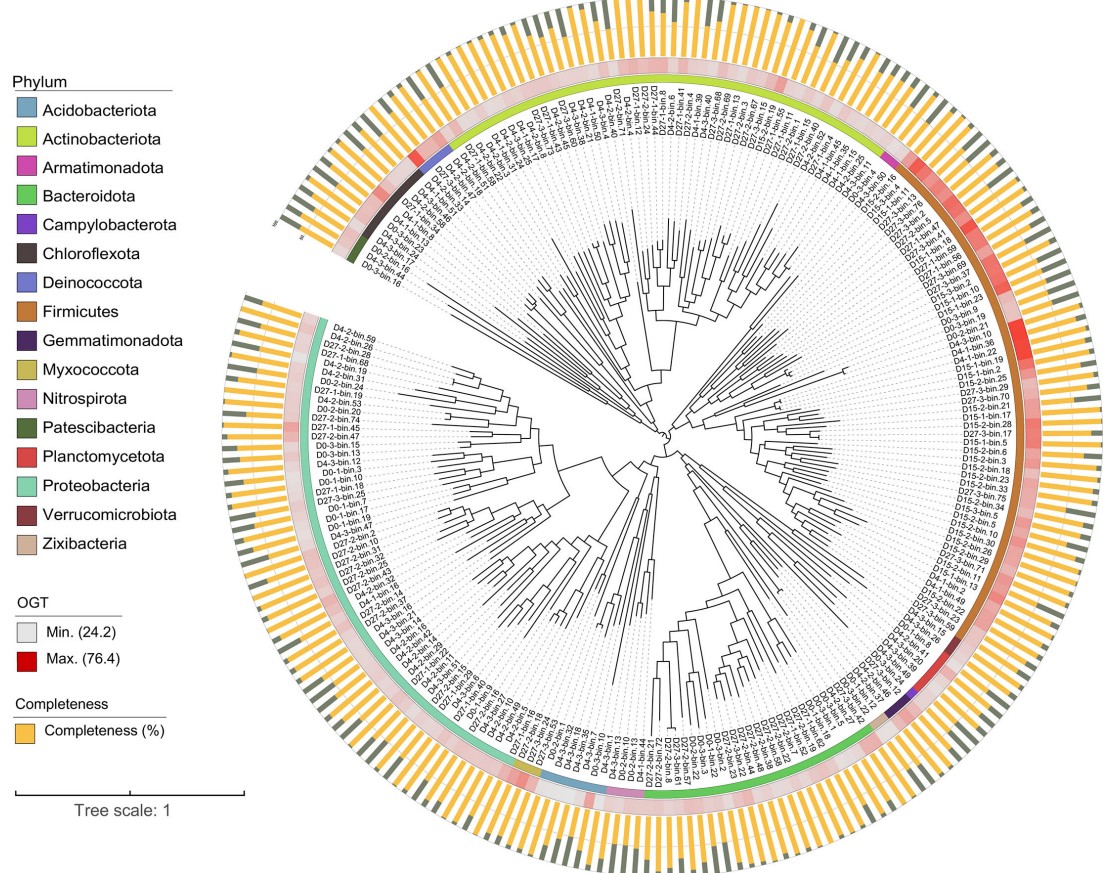

**FIG 3** Overview of compost bacterial community based on recovered 227 MAGs from metagenomic binning during the HTC. Inner circles are colored by the taxonomic classification of MAGs at the phylum level, while the second inner circle color indicates the OGT of MAGs (legend on the right). The bar graphs shown on the outer circle represent the genomic completeness (yellow) and contamination (gray) of MAGs determined by CheckM.

addition, a *Thermus thermophilus* MAG (D4-2-bin.33, Deinococcota phylum) with an OGT of 70°C was identified, consistent with experimental data for *T. thermophilus* isolates (43).

To track changes in MAG abundance throughout the composting process, we quantified MAG abundance at each composting stage. Thermophilic and mesophilic MAGs displayed distinct temporal patterns. Thermophilic MAGs were enriched during the high-temperature phase, whereas mesophilic MAGs dominated the initial and maturation phases (Fig. 4). The relative abundance of thermophilic MAGs increased dramatically—by 43-fold—from 1.8% at D0 to 82.2% at D15, then decreased to 10.3% by D27 (Fig. 4a). In contrast, mesophilic MAG abundance decreased twofold by D15 compared to D0. While 16S rRNA sequencing has previously captured bacterial community shifts during HTC (6, 8), this is the first genome-resolved analysis of thermophilic bacterial community dynamics during composting. During the hyperthermophilic and thermophilic phases (D4 to D15), the dominant thermophilic MAGs—*T. thermophilus*, *Planifilum fulgidum*, and *Thermaerobacter* spp.—accounted for more than 87% of total MAG abundance. The abundance of *Thermus* and *Planifilum* based on 16S rRNA sequencing was positively correlated with composting temperature (Fig. S8), further supporting their role as key thermophiles in heat generation during composting.

Transcriptional activity of MAGs also mirrored temperature changes during HTC (Fig. 4b). At D15, thermophilic MAGs exhibited a 33.9-fold increase in transcriptional activity compared to D0, while mesophilic MAGs showed a 9-fold decrease. Three MAGs—D27-1-bin.59 (*P. fulgidum*, OGT 58.9°C), D15-1-bin.23 (*Thermaerobacter*, OGT 68.1°C), and D15-3-bin.4 (*T. thermophilus*, OGT 69.8°C)—were the most transcriptionally active at D15,

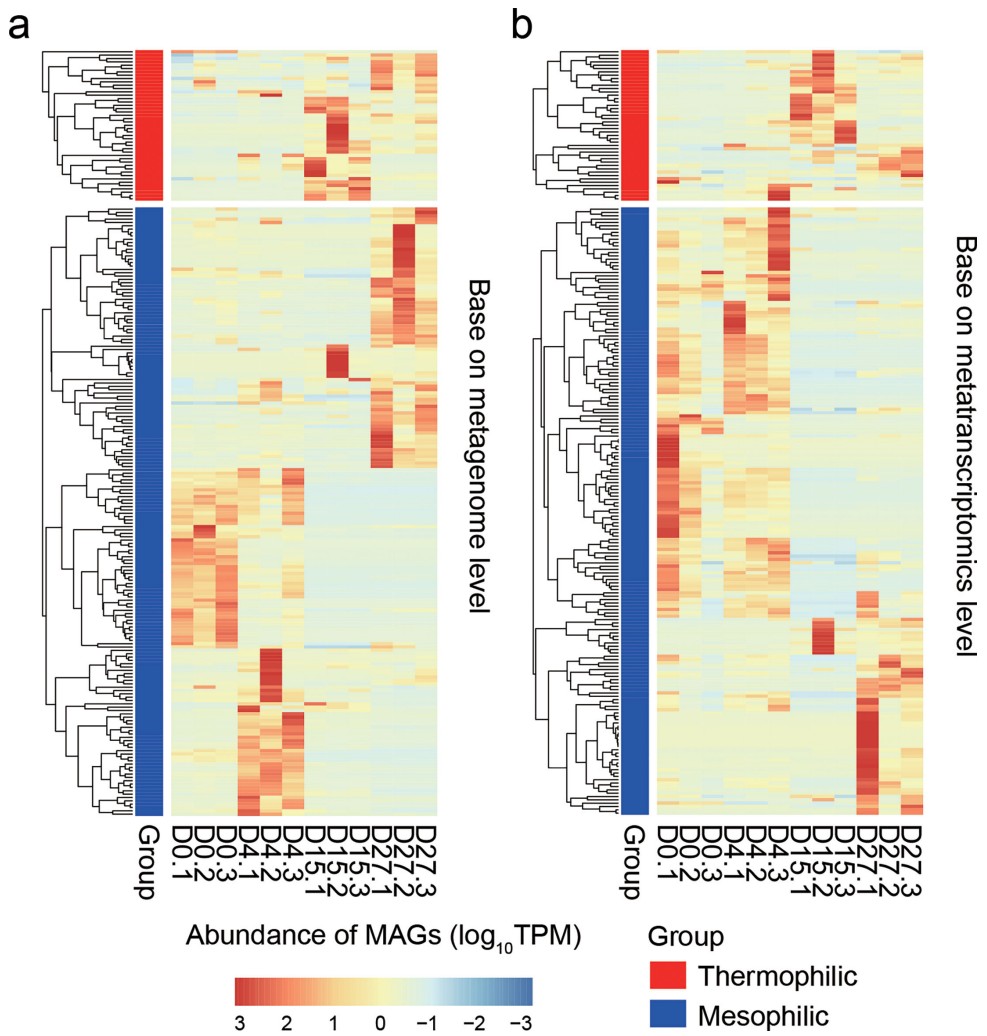

**FIG 4** Comparison of genomic abundance and metabolic activity of mesophilic and thermophilic bacteria (MAGs) during HTC based on the transcriptional activity of functional genes. (a) Heatmap showing changes in the genomic abundance of mesophilic (OGT < 45°C) and thermophilic (OGT > 45°C) bacteria during HTC. (b) Heatmap showing changes in the transcriptional activity of functional genes from mesophilic and thermophilic MAGs during HTC. In both panels, D0, D4, D15, and D27 refer to samples collected on days 0 (initial phase), 4 (hyperthermophilic phase), 15 (thermophilic phase), and 27 (maturation phase) of the composting process.

contributing 73.8% of total activity among thermophilic MAGs. These results suggest that these three thermophiles are key contributors to heat production during composting, in agreement with our previous findings (11).

Interestingly, while other thermophilic MAGs had high predicted OGTs, their transcriptional activity was relatively low. For example, a *Calditerricola* MAG (D4-1-bin.36, OGT 76.4°C) showed minimal transcriptional activity throughout composting. Taken together, these findings indicate that a small number of key thermophilic bacteria, particularly *T. thermophilus*, *P. fulgidum*, and *Thermaerobacter*, play a central role in microbial heat production during HTC.

## Genomic survival strategies of thermophilic bacteria during the hyperthermophilic phase of HTC

To investigate the metabolic roles of MAGs during HTC composting, we analyzed the functional gene profiles of both thermophilic and mesophilic MAGs. All MAGs encoded

essential metabolic pathways involved in carbon, nitrogen, energy, fatty acid, and nucleotide metabolism, as well as stress responses (Table S4). Each MAG contained a large number of carbohydrate-active enzyme (CAZyme) genes—including glycoside hydrolases, glycosyltransferases, carbohydrate esterases (CEs), and auxiliary activity enzymes (AAs)—with an average of 111 CAZyme genes per MAG, underscoring their role in carbohydrate decomposition (Table S5). Furthermore, 70% of MAGs carried genes involved in nitrogen cycling processes such as denitrification, anammox, nitrite oxidation, nitrate reduction, and nitrogen fixation, all of which are critical to organic matter decomposition and nutrient transformation during composting (44).

Genomic analysis revealed that mesophilic MAGs harbor significantly more genes related to carbon and nitrogen cycling than thermophilic MAGs ($P < 0.01$; Fig. S9), suggesting that mesophiles likely have greater potential to drive carbon and nitrogen turnover during the HTC composting process compared to thermophiles. In contrast, dominant thermophilic lineages such as *Thermus* and *Planifilum* exhibited markedly elevated transcriptional activity during the high-temperature phase (Fig. S8). This pattern may reflect a survival strategy in which these thermophilic populations prioritize thermal adaptation over nutrient metabolism to maintain ecological competitiveness. Consistent with this interpretation, previous findings suggest that thermophiles may have streamlined their genomes—losing certain metabolic pathways—to better adapt to high-temperature environments (45). Supporting this, thermophilic MAGs had significantly smaller genome sizes (average 2.2 Mbp) compared to mesophilic MAGs (2.9 Mbp), with a strong negative correlation ($P < 0.001$) between genome size and OGT across all MAGs (Fig. S10), aligning with prior reports that genome reduction can enhance thermotolerance by reducing energy demands (46).

To further explore thermophile survival strategies, we examined 19 high-quality thermophilic MAGs (completeness > 90%, contamination < 5%; Fig. S11). All thermophilic MAGs encoded cytochrome c oxidase, indicating aerobic respiratory capabilities. Additionally, cbb3-type cytochrome oxidases—adapted for microaerobic or anaerobic conditions—were detected in seven MAGs (47), suggesting their ability to persist in low-oxygen microenvironments within the compost matrix. All thermophilic MAGs encoded DNA repair genes such as *radA* and *recN*, and six MAGs possessed heat shock proteins (HSPs; Fig. 5a), known for conferring resistance to thermal stress (17, 46). Reverse gyrase, a hallmark of extreme thermophiles, was exclusively detected in *T. thermophilus* (D4-2-bin.33) (48). Motility genes were also prevalent, with 13 MAGs encoding flagellar systems and seven encoding type IV pili, possibly facilitating migration to thermally favorable niches. Although not expressed under ambient conditions, these genes were upregulated during the thermophilic phase, as shown in Fig. 5a. Spore formation genes were widespread among MAGs (Table S6), supporting the importance of endospores for survival in extreme environments. Moreover, many thermophilic MAGs encoded genes for the biosynthesis of compatible solutes such as trehalose, polyphosphate, ectoine, and glycine betaine—well-established osmoprotectants that enhance stress resilience in extremophiles (49, 50).

Metatranscriptomic data validated these genomic predictions. Expression levels of key thermal response genes—including *hsp20*, *reverse gyrase* (*rgy*), and *fliE* (a flagellar complex gene)—were significantly upregulated at D4 and D15, compared to the ambient phase at D0 (Fig. 5b). Among the 19 thermophilic MAGs, four (*Planifilum fulgidum*, *Sphaerobacter thermophilus*, *Thermaerobacter*, and *T. thermophilus*) showed the highest abundance and transcriptional activity at D15, collectively contributing 35.4% of total transcriptional activity during this high-temperature stage. Metabolic pathway reconstruction revealed these key taxa encoded multiple adaptive traits, including Hsp20, HSPA9, and robust DNA repair systems, enabling central metabolic processes—glycolysis, the methylerythritol 4-phosphate (MEP) pathway, the TCA cycle, fatty acid metabolism, and energy metabolism—to proceed under extreme heat (Fig. 6). Interestingly, three of the six known autotrophic carbon fixation pathways—the Calvin-Benson-Bassham (CBB) cycle, the reverse tricarboxylic acid (rTCA) cycle, and the Wood–Ljungdahl

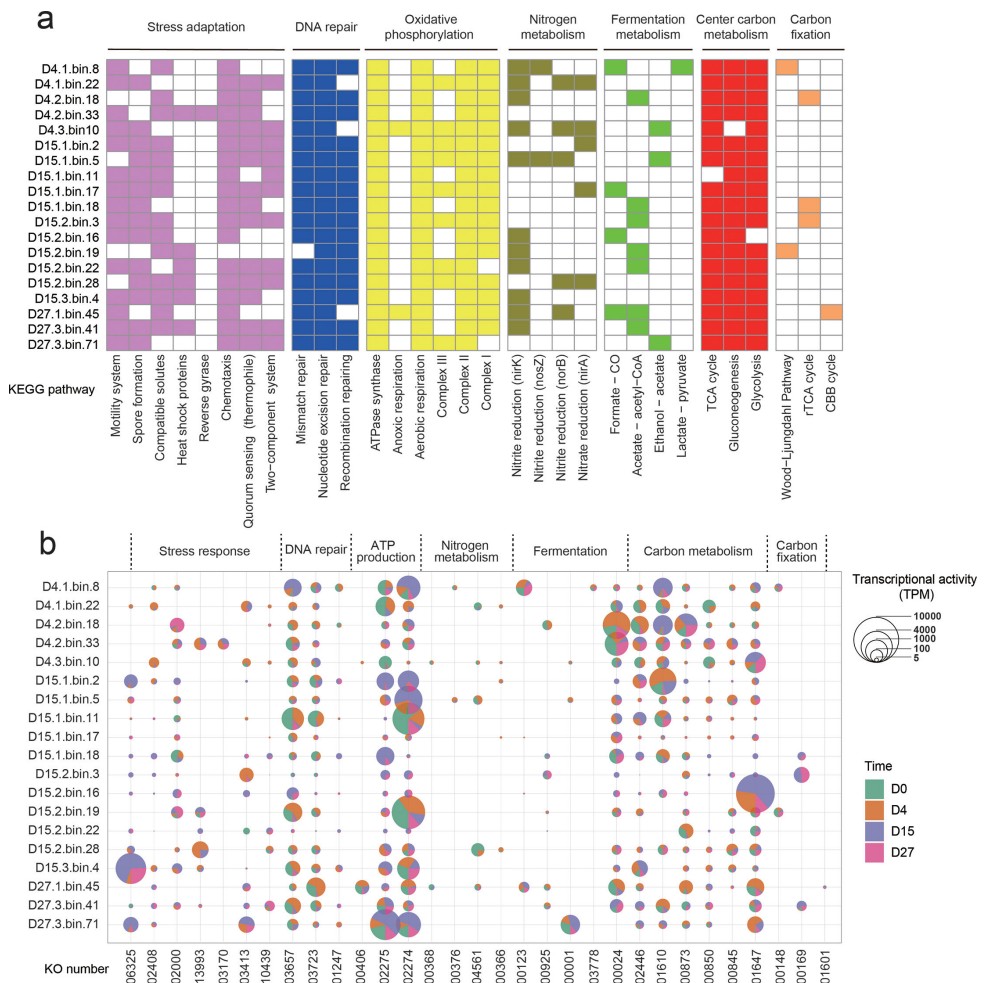

**FIG 5** Characterization of core functional genes involved in carbon fixation, energy and carbon metabolism, nitrogen metabolism, fermentation, and stress response in refined thermophilic MAGs. (a) The presence of specific genes or pathways is indicated by colored boxes for each thermophilic MAG. Pathways such as glycolysis, gluconeogenesis, and the TCA cycle are considered present if ≥90% of the corresponding genes are detected. Fermentation metabolism is indicated by the presence of key dehydrogenase genes. Carbon fixation metabolism is considered present if at least two complete key enzymatic pathways are identified within a MAG. Detailed gene information used to define each metabolic function is provided in Table S6. (b) Normalized transcript abundance of key genes related to carbon metabolism, nitrogen metabolism, ATP production, respiration, and stress adaptation in refined thermophilic MAGs across composting time points (D0, D4, D15, and D27) during HTC. Expression levels are presented as TPM. Blank sections in the pie charts represent undetected gene activity in the metatranscriptomes.

(WL) pathway—were partially encoded in several thermophilic MAGs (Fig. 5a). For instance, *D27-1-bin.45* (*Nitrococcales*) encoded nearly the complete CBB cycle, including key genes for ribulose-1,5-bisphosphate carboxylase and phosphoribulokinaseThe WL pathway, considered among the most ancient carbon fixation mechanisms, was partially present in *D4-1-bin.8* (*Thermobifida fusca*) and *D15-2-bin.19* (*S. thermophilus*), missing only the methyl branch component (e.g., acetyl-CoA synthase), possibly due to assembly gaps (51). Similarly, the rTCA pathway was partially encoded in *D4-2-bin.18* (*Trueperaceae*) and *D15-2-bin.3* (*Paenibacillales*), including key enzymes like *porA*, *ppsA*, and *korA*. To our knowledge, this represents the first genomic evidence of CO$_2$ fixation in thermophilic composting systems, though the contribution of CO$_2$ fixation to thermal adaptation remains speculative. Notably, recent studies have revealed a broader distribution of autotrophic carbon fixation across microbial taxa than previously recognized (52).

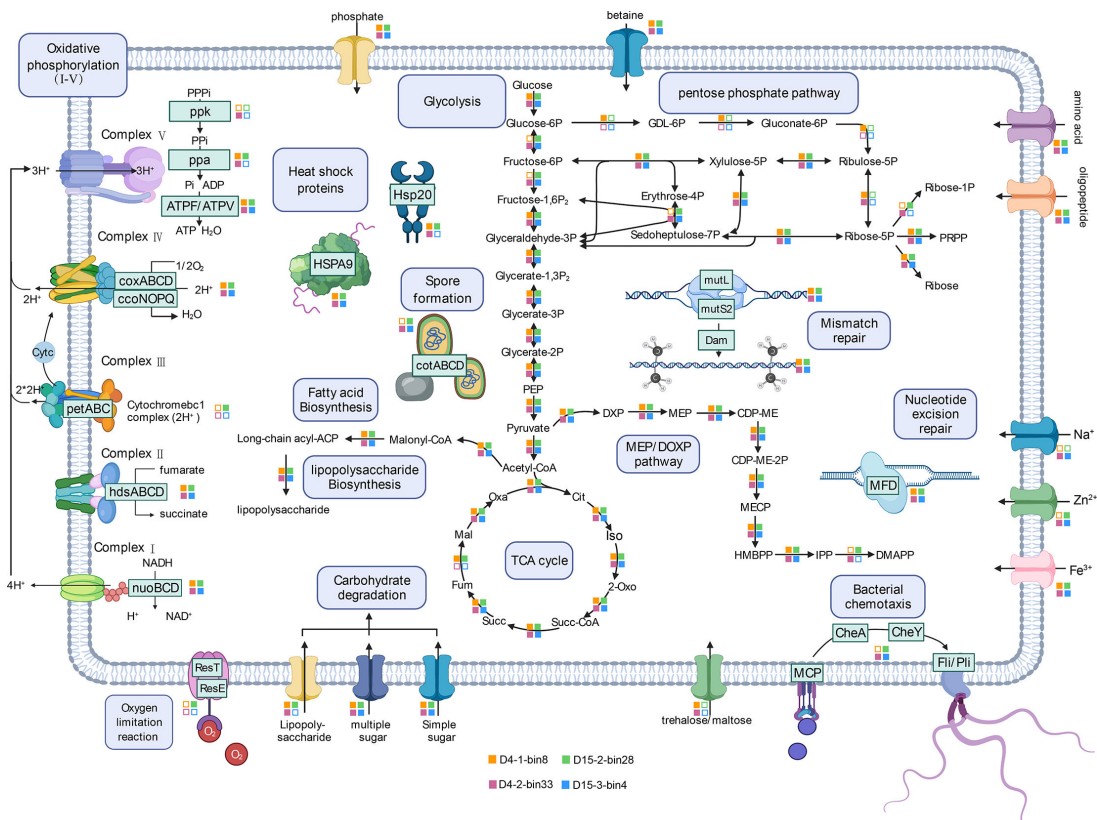

**FIG 6** Reconstruction of the metabolic capabilities of the four key thermophilic MAGs with the highest transcriptional activity during HTC composting (D4-1-bin.8, D4-2-bin.33, D15-2-bin.28, and D15-3-bin.4). The figure illustrates the presence or absence of genes involved in glycolysis, the MEP pathway, the tricarboxylic acid (TCA) cycle, fatty acid metabolism, energy metabolism, membrane transporters, and other stress adaptation mechanisms. Colored squares indicate the presence of corresponding enzymes in each MAG, while colorless squares indicate their absence. Genes corresponding to each metabolic pathway depicted in the figure are listed in Table S5.

During HTC, thermophilic MAGs expressed high levels of genes related to core metabolic functions—central carbon metabolism, and energy generation, DNA repair, and stress response—while transcripts associated with $CO_2$ fixation, nitrogen cycling, and fermentation remained relatively low (Fig. 5b). These results suggest that thermophilic bacteria rely on tightly regulated genetic and transcriptional programs to survive and function under extreme HTC conditions. In summary, this study offers new insights into the metabolic activity and adaptive strategies of thermophilic bacteria in HTC composting. Despite the presence of numerous thermophilic taxa, only a few highly active strains dominate during the hyperthermophilic phase, highlighting their central role in driving both heat production and functional processes in high-temperature compost ecosystems.

### ACKNOWLEDGMENTS

This work was supported by the National Key Research and Development Program of China (2023YFD1702200), the National Natural Science Foundation of China (42277357), and the science and technology innovation Program of Hunan Province (2022RC3057).

C.L., Y.H., Q.-E.Y., H.Z., D.Z., C.A., X.T., H.L., and Z.T. analyzed the majority of the laboratory experiment data, while C.L. and Q.-E.Y. conducted and analyzed most of the composting experiment data. H.Z., S.T., C.A., and C.L. performed most of the statistical analyses. C.L., X.T., and H.L. designed the experiments, contributed intellectual input, and assisted with the sequencing data analyses. H.Z. and C.L. wrote the manuscript with

input and revisions from all co-authors. V.-P.F., Z.T., and Q.-E.Y provided feedback and revised the manuscript. H.L. and S.Z. supervised the study.

The authors declare no competing financial interest.

## AUTHOR AFFILIATIONS

[1]Fujian Provincial Key Laboratory of Soil Environmental Health and Regulation, College of Resources and Environment, Fujian Agriculture and Forestry University, Fuzhou, China
[2]Research Center for Eco-Environmental Engineering, Dongguan University of Technology, Dongguan, China
[3]Yuelushan Laboratory, Changsha, China
[4]Department of Microbiology, University of Helsinki, Helsinki, Finland

## AUTHOR ORCIDs

Chen Liu  http://orcid.org/0009-0009-1707-574X
Qiu-e Yang  http://orcid.org/0000-0002-0552-7763
Hanpeng Liao  http://orcid.org/0000-0001-7539-2668
Shungui Zhou  http://orcid.org/0000-0003-0899-4225

## FUNDING

| Funder | Grant(s) | Author(s) |
|---|---|---|
| National Key Research and Development Program of China | 2023YFD1702200 | Hanpeng Liao |
| | | Shungui Zhou |
| National Natural Science Foundation of China | 42277357 | Hanpeng Liao |
| Science and technology innovation Program of Hunan Province | 2022RC3057 | Shiyong Tan |
| Yuelushan Laboratory Breeding Program | YLS-2025-ZY02039 | Shiyong Tan |

## AUTHOR CONTRIBUTIONS

Chen Liu, Conceptualization, Data curation, Formal analysis, Investigation, Writing – original draft | Yuqi He, Conceptualization, Formal analysis, Software, Writing – original draft | Hongbo Zhang, Formal analysis, Software | Dong Zhang, visualization, Writing – review and editing | Chaofan Ai, Methodology, Validation | Xiang Tang, Methodology, Writing – review and editing | Qiu-e Yang, Resources, Writing – review and editing | Zhen Yu, Formal analysis, Resources | Shiyong Tan, Funding acquisition, Resources | Ville-Petri Friman, Investigation, Writing – review and editing | Hanpeng Liao, Funding acquisition, Project administration, Resources, Supervision, Writing – review and editing | Shungui Zhou, Funding acquisition, Resources, Writing – review and editing

## DATA AVAILABILITY

Raw read data from both amplicon and shotgun sequencing in this study are publicly available in the NCBI Sequence Read Archive (SRA) under the accession numbers PRJNA861164, PRJNA861429, and PRJNA861433.

## ADDITIONAL FILES

The following material is available online.

### Supplemental Material

**Supplemental material (mSystems00956-25-s0001.docx).** Supplemental text and figures.
**Supplemental tables (mSystems00956-25-s0002.xlsx).** Tables S1 to S6.

Open Peer Review

**PEER REVIEW HISTORY (review-history.pdf).** An accounting of the reviewer comments and feedback.

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
