## [Reviewer comments · mSystems]

Metabolic activity and survival strategies of thermophilic microbiomes during hyperthermophilic composting

Chen Liu, Yuqi He, Hongbo Zhang, Dong Zhang, Chaofan Ai, Xiang Tang, Qiu Yang, Zhen Yu, Shiyong Tan, Ville-Petri Friman, Hanpeng Liao, and Shun-Gui Zhou

Corresponding Author(s): Hanpeng Liao, Fujian Agriculture and Forestry University

Review Timeline:

Submission Date:	June 26, 2025
Editorial Decision:	August 2, 2025
Revision Received:	August 19, 2025
Editorial Decision:	September 8, 2025
Revision Received:	September 10, 2025
Accepted:	September 15, 2025

Editor: Xue Guo

Reviewer(s): Disclosure of reviewer identity is with reference to reviewer comments included in decision letter(s). The following individuals involved in review of your submission have agreed to reveal their identity: Linwei Wu (Reviewer #1)

Transaction Report:

DOI: <https://doi.org/10.1128/msystems.00956-25>

Re: mSystems00956-25 (**Metabolic activity and survival strategies of thermophilic microbiomes during hyperthermophilic composting**)

Dear Prof. Hanpeng Liao:

Both reviewers think that the study is highly significant and the manuscript is excellently presented. However, minor revisions need to be made in certain sections. Please revise the manuscript accordingly by addressing the reviewers' comments provided below.

Revision Guidelines

Sincerely,
Xue Guo
Editor
mSystems

Reviewer #1 (Comments for the Author):

This study provides a comprehensive exploration of thermophilic microbial communities in HTC, combining metagenomics and metatranscriptomics to reveal their metabolic activity and survival strategies. The results successfully identify key thermophilic taxa (e.g., *Thermus thermophilus*, *Planifilum fulgidum*) that drive heat production. The multi-omics approach offers valuable genome-resolved insights into microbial function under extreme conditions, with potential applications for optimizing waste treatment. However, some discussions lack sufficient rigor and require substantial revision. Additionally, all citations should be placed within sentences rather than after them. Below are detailed comments:

Line 39: Change to "Keywords:"

Introduction:

Lines 79-81: Revise the sentence due to two consecutive dashes

Lines 81-83: These two sentences don't demonstrate an example relationship - either adjust the logic or combine into one sentence

Lines 107-108: It is a rather uninteresting hypothesis that could potentially be deleted

Lines 109-110: "Diverse adaptive mechanisms" should be expanded with specific examples. The introduction needs 1-2 sentences to set this up.

Materials and Methods:

Standardize company abbreviations to either "Co., Ltd." or "Co, Ltd"

Lines 121-122: Some redundancy with lines 117-119 - consider deletion

Lines 144-151: Methods for physicochemical measurements should be listed in the same order as mentioned in the previous sentence

Line 166: Add sequencing depth information for metagenomic data

Line 237: PERMANOVA (adonis) and ANOSIM are different methods. Potential confusion here needs addressing.

Results and Discussion:

Line 245: Redundant explanation of HTC

Lines 259-261: Current Figure 1d shows Weighted UniFrac (Y-axis) significantly changing with temperature (X-axis)

Lines 264-266: The current content doesn't support the stated conclusion

Lines 280-282: For rigorous ARG results, recommend using dedicated ARG databases for annotation

Lines 312-313: This paragraph's conclusion isn't sufficiently supported

Lines 321-322: Italics formatting didn't need for phyla

Lines 324-326: The statement about MAGs not being classifiable by GTDB-TK doesn't directly relate to <1% of microbes being cultured. Revise this argument.

Lines 347-349: "Thermophilic phase (D4 to D15)" contradicts the phase classification in Figure captions (lines 570-573).

Lines 379-380: Figure S8 doesn't conclusively support this claim

Figure Captions:

Lines 570-573: Clarify how days 10-15 are classified in the phase system

Reviewer #2 (Comments for the Author):

The manuscript explores the composition of hyperthermophilic composting (HTC) via a multi-omics approach. The study supports the utilization of treating organic solid waste by using specialized microbial communities, which the authors have profiled and presented well in the paper. The overall content is well within the scope of mSystems. I just have a few clarifications and questions, listed below:

1) Is temperature the only key factor in shaping the microbial composition of HTC? Are organic matter and nutrient cycling during the phases also involved in HTC, particularly the hyperthermophilic and maturation phases? Do they also play roles in shaping the microbial composition (at least for the key abundant species)? I recommend that the authors discuss further this further.

2) In "Materials and Methods: Hyperthermophilic composting experiments", the authors wrote "four" distinct temperature phases but only enumerated three. Please clarify.

3) Which 16S rDNA region was sequenced? It is beneficial that the authors identify and support such a choice.

4) In the "Results and Discussion: The succession of the bacterial community drives heat production during HTC", the authors wrote that the temperature reached a peak of 94dC within 24 h. However, in the Figure, it took approximately 2.5 days to peak at 94dC. Please clarify.

Responses to the reviewer's comments

Editors' comments to the author:

Both reviewers think that the study is highly significant and the manuscript is excellently presented. However, minor revisions need to be made in certain sections.

Please revise the manuscript accordingly by addressing the reviewers' comments provided below.

Reply: We sincerely thank you for handling our manuscript and for the constructive feedback from both reviewers. We have revised the manuscript accordingly, addressing all comments point by point. Changes are highlighted in the revised version. We hope the revised manuscript meets the requirements for publication in mSystems.

Reviewer #1 (Comments to the Author):

*This study provides a comprehensive exploration of thermophilic microbial communities in HTC, combining metagenomics and metatranscriptomics to reveal their metabolic activity and survival strategies. The results successfully identify key thermophilic taxa (e.g., *Thermus thermophilus*, *Planifilum fulgidum*) that drive heat production. The multi-omics approach offers valuable genome-resolved insights into microbial function under extreme conditions, with potential applications for optimizing waste treatment. However, some discussions lack sufficient rigor and require substantial revision. Additionally, all citations should be placed within sentences rather than after them. Below are detailed comments:*

R (Reply): We sincerely appreciate your thorough and constructive review. We are encouraged by your positive assessment of the significance and quality of our study, particularly your recognition of the multi-omics approach and its implications for understanding microbial function in extreme environments. We have carefully addressed each of your comments and revised the manuscript accordingly to improve

the clarity, rigor, and logical consistency of our arguments. In particular, we have strengthened relevant discussions and ensured that all citations are now placed appropriately within sentences, as suggested. Our detailed response to each of your points was provided below.

Major comments:

Q1 (Question 1): Line 39: Change to "Keywords:"

R1: Revised as suggested (line 39).

Q2: Lines 79-81: Revise the sentence due to two consecutive dashes.

R2: Revised as suggested to improve clarity and avoid ambiguity.

Revised in text (lines 79–80): “However, in 2008, hyperthermophilic composting (HTC) technology was developed, achieving temperatures above 90°C without external heating, which is 20–30°C higher than conventional composting⁹.”

Q3: Lines 81-83: These two sentences don't demonstrate an example relationship - either adjust the logic or combine into one sentence

R3: We have revised this part to improve the logical connection between the sentences.

Revised in text (Lines 82–84): “HTC can maintain high microbial activity under extreme thermal stress, thereby enhancing organic matter degradation and humus formation^{8, 12}.”

Q4: Lines 107-108: It is a rather uninteresting hypothesis that could potentially be deleted

R4: We deleted the original scientific hypothesis (line 104).

Q5: Lines 109-110: “Diverse adaptive mechanisms” should be expanded with specific examples. The introduction needs 1-2 sentences to set this up.

R5: We have revised the hypothesis to include representative examples of adaptive

strategies.

Revised in text (Lines 105–112): “(2) thermophilic bacteria employ a range of physiological and molecular strategies to cope with extreme thermal stress during HTC, such as sporulation, shifts in cellular behavior, and activation of stress response pathways. From the HTC composting process, 517 medium-quality MAGs were recovered through metagenomic binning. Growth temperature predictions identified 45 thermophilic MAGs likely contributing to heat production and metabolism. Genome functional annotation and metatranscriptomic analyses revealed that these thermophiles employed diverse survival and adaptation strategies under high-temperature conditions, involving physiological adjustments and the induction of thermal resistance mechanisms.”

Q6: Standardize company abbreviations to either "Co., Ltd." or "Co, Ltd"

R6: We have revised all company names to the format “Co., Ltd.” throughout the manuscript (lines 120, 128).

Q7: Lines 121-122: Some redundancy with lines 117-119 - consider deletion

R7: We have deleted the duplicate sentences to improve clarity.

Revised in text (lines 119–123): “In this study, full-scale hyperthermophilic composting (HTC) experiments were conducted using composting material at GeoGreen Innotech Co., Ltd., in Beijing, China, as previously described²¹. The raw materials were also sourced from the same facility, which consisted of dehydrated sewage sludge (80% water content), rice husk (13% water content), and composting by-products (40% water content) under the 1:1:4 ratio (by weight).”

Q8: Lines 144-151: Methods for physicochemical measurements should be listed in the same order as mentioned in the previous sentence

R8: The order of physicochemical parameters has been revised to match the order of the corresponding methods.

Revised in text (lines 141–144): “The physicochemical properties of the compost,

including water content (WC), ammonium (NH₄⁺), nitrate (NO₃⁻), total organic carbon (TOC), dissolved total nitrogen (DN), inorganic carbon (IC), total carbon (TC), total nitrogen (TN), electrical conductivity (EC), and pH, were measured following previously described methods²². Water content was determined by drying compost samples at 105 °C for 24 hours using the drying oven. Ammonium and nitrate were extracted using 2 M KCl and measured with a continuous-flow autoanalyzer after a 10-fold dilution (FlowSys, Systea, Rome, Italy). TOC, DN, and IC were extracted using deionized water at a 1:10 (v:v) ratio and analyzed with an automated TOC analyzer (Shimadzu TOC-L CPH, Kyoto, Japan). TC and TN were measured via dry combustion using an Elementar CHNS analyzer (Vario MAX cube, Hanau, Germany), and the C/N ratio was calculated accordingly. Electrical conductivity and pH were measured using a pH meter with deionized water at a 1:10 (v:v) ratio for the extraction.”

Q9: Line 166: Add sequencing depth information for metagenomic data

R9: We have added a summary of sequencing metagenomic data in the Methods and Materials section including sequencing depth, raw read counts, base yield, quality scores, GC content, and filtering efficiency in Supplementary Table S1.

Revised in text (lines 174–175): “Shotgun metagenomic sequencing generated approximately 60–120 million raw reads per sample, corresponding to 7.3–17.7 Gbp per library (Table S1).”

Q10 : Line 237: PERMANOVA (adonis) and ANOSIM are different methods. Potential confusion here needs addressing.

R10: We apologize for the error in the text. We have corrected this error in revised version.

Revised in text (lines 239–240): “Differences in microbial community composition across composting stages were assessed using non-parametric PERMANOVA (adonis function, 999 permutations) based on a distance matrix.”

Q11: Line 245: Redundant explanation of HTC

R11: Removed the redundant explanation of HTC (line 246).

Q12: Lines 259-261: Current Figure 1d shows Weighted UniFrac (Y-axis) significantly changing with temperature (X-axis)

R12: We have revised the description to better align with the relationship shown in Fig. 1d and clearly clarified that the association is based on changes in composting temperature.

Revised in text (lines 260-262): “Microbial community dissimilarities (based on Weighted UniFrac distance) were strongly correlated with changes in composting temperature ($p < 0.001$), indicating that bacterial succession is a key driver of heat production during HTC (Fig. 1d).”

Q13: Lines 264-266: The current content doesn't support the stated conclusion

R13: We apologize for this misunderstanding. The last sentence was intended to highlight the advantages of high temperatures in the HTC composting process, rather than to summarize the preceding results. We have now revised the text to include a more relevant summary at the end of this section.

Revised in text (lines 271–275): “Our HTC used thermophiles to achieve and sustain extreme thermophilic conditions, thereby enhancing organic matter degradation and pathogen inactivation compared to traditional composting^{35, 39}. Taken together, these results indicate that the bacterial community structure undergoes dramatic shifts throughout the stages of HTC composting, with thermophilic bacteria playing a crucial role in producing extreme high temperatures.”

Q14: Lines 280-282: For rigorous ARG results, recommend using dedicated ARG databases for annotation

R14: Thank you for your suggestion. We would like to clarify that the results presented here focus on changes in KEGG pathway genes related to human diseases, rather than on antibiotic resistance genes (ARGs) themselves. While pathogens and

ARGs are discussed as potential factors influencing the observed shifts in human disease pathway genes, they are not the results of this analysis. We have revised the sentence to clarify this.

Revised in text (Lines 288–291): “Interestingly, pathways related to human diseases showed a significant reduction in abundance during the thermophilic phase compared to the initial phase (Fig. S3), likely due to the efficient inactivation of pathogens and antibiotic resistance genes at temperatures exceeding 80°C⁶.”

Q15: Lines 312-313: This paragraph's conclusion isn't sufficiently supported

R15: We have revised the summary conclusion of this section to offer a more precise and cautious interpretation that accurately reflects the observed association between microbial metabolic activity and temperature dynamics during the composting process.

Revised in text (Lines 318–321): “Taken together, the observed shifts in bacterial metabolic activity across the composting phases closely reflect the temperature dynamics throughout the HTC process. Notably, a small group of key thermophilic bacteria dominate heat production and metabolic activity during the high-temperature phase.”

Q16: Lines 321-322: Italics formatting didn't need for phyla

R16: We have corrected it accordingly (Lines 329–330).

Q17: Lines 324-326: The statement about MAGs not being classifiable by GTDB-TK doesn't directly relate to <1% of microbes being cultured. Revise this argument.

R17: Thanks. We have provided a more accurate and clearer statement on this.

Revised in text (lines 332–335): “It is worth noting that 195 MAGs could not be classified to any known genus using the current GTDB-Tk database, indicating a substantial number of unknown bacterial groups in HTC. Compared with traditional isolation and culture methods, culture-independent metagenomics provides genomic insights into a vast diversity of uncultivated microbiomes.”

Q18: Lines 347-349: "Thermophilic phase (D4 to D15)" contradicts the phase classification in Figure captions (lines 570-573).

R18: We apologize for this inadvertent error. We have revised the phase descriptions in the relevant sentence to match the composting stage definitions provided in the Materials and Methods section, ensuring consistency across the manuscript.

Revised text:

Lines 130–130: "the initial phase (Day 0–1), the hyperthermophilic phase (days 2–14), thermophilic phase (days 15–26), and maturation phase (days 27–45)."

Line 356: "During the hyperthermophilic and thermophilic phases (D4 to D15), the dominant thermophilic MAGs—*Thermus thermophilus*, *Planifilum fulgidum*, and *Thermaerobacter* spp.—accounted for more than 87% of total MAG abundance."

Lines 586–590: "In panels a and b, P0, P1, P2, and P3 represent the initial, hyperthermophilic, thermophilic, and maturation phases, respectively. The initial phase includes samples from Day 0; the hyperthermophilic phase includes samples from Days 4 and 9; the thermophilic phase includes samples from Days 15 and 21; and the maturation phase includes samples from Days 27 and 45. Each time point comprises five biological replicates."

Q19: Lines 379-380: Figure S8 doesn't conclusively support this claim

R19: Sorry. We have revised the expression of this claim to offer a more precise and cautious interpretation.

Revised in text (lines 386–390): "Genomic analysis revealed that mesophilic MAGs harbor significantly more genes related to carbon and nitrogen cycling than thermophilic MAGs ($p < 0.01$, Fig. S9), suggesting that mesophiles likely have greater potential to drive carbon and nitrogen turnover during the HTC composting process compared to thermophiles. In contrast, thermophilic MAGs showed a reduced representation of nutrient cycling function, potentially reflecting adaptations to extreme thermal stress."

Q20: *Lines 570-573: Clarify how days 10-15 are classified in the phase system*

R20: We sincerely apologize for the incorrect statement. The times mentioned refer to sampling time points, not to the division of the four composting stages. Based on temperature dynamics, four distinct composting phases were identified: the initial phase (Days 0–1), the hyperthermophilic phase (Days 2–14), the thermophilic phase (Days 15–26), and the maturation phase (Days 27–45). The text has been revised accordingly to correct this expression.

Revised in text (Lines 586–590): “In panels a and b, P0, P1, P2, and P3 represent the initial, hyperthermophilic, thermophilic, and maturation phases, respectively. The initial phase includes samples from Day 0; the hyperthermophilic phase includes samples from Days 4 and 9; the thermophilic phase includes samples from Days 15 and 21; and the maturation phase includes samples from Days 27 and 45. Each time point comprises five biological replicates.”

Reviewer #2 (Comments for the Author):

The manuscript explores the composition of hyperthermophilic composting (HTC) via a multi-omics approach. The study supports the utilization of treating organic solid waste by using specialized microbial communities, which the authors have profiled and presented well in the paper. The overall content is well within the scope of mSystems. I just have a few clarifications and questions, listed below:

R: We sincerely thank you for the positive comments and for recognizing our study’s contribution to advancing the understanding of hyperthermophilic composting through multi-omics approaches. For the detail comments, we have addressed each of them in detail below.

Major comments:

Q1 (Question 1): *Is temperature the only key factor in shaping the microbial composition of HTC? Are organic detaileded and nutrient cycling during the phases also involved in HTC, particularly the hyperthermophilic and maturation phases? Do they also play roles in shaping the microbial composition (at least for the key*

abundant species)? I recommend that the authors discuss further this further.

R1: We fully agree with your opinions. Besides composting temperature, factors such as organic content and nutrient availability also influence microbial community succession during the composting process. Accordingly, we have added an analysis of the relationships between additional physicochemical properties of the compost and microbial community succession in the Results section (lines 264–267). Furthermore, we have supplemented the Discussion to address these factors (lines 267–269).

Revised in text (Lines 262–267): “Mantel tests based on Bray–Curtis dissimilarities further showed that other physicochemical parameters—such as nutrient content, pH, and moisture—also exhibited significant associations with bacterial community composition (Fig. S2). These results indicate that multiple factors influence microbial succession during HTC. Nevertheless, temperature remains the most direct determinant of microbial survival and metabolic activity in composting systems, especially during the hyperthermophilic phase³⁵.”

Q2: *In "Materials and Methods: Hyperthermophilic composting experiments", the authors wrote "four" distinct temperature phases but only enumerated three. Please clarify.*

R2: The HTC process in our study comprised four distinct temperature phases: the initial phase (Day 0–1), the hyperthermophilic phase (Days 2–14), the thermophilic phase (Days 15–26), and the maturation phase (Days 27–45). We have corrected the Materials and Methods section to ensure that all four phases are enumerated consistently in the text.

Revised in text (Lines 130–132): “the initial phase (Day 0–1), the hyperthermophilic phase (days 2–14), thermophilic phase (days 15–26), and maturation phase (days 27–45).”

Q3: *Which 16S rDNA region was sequenced? It is beneficial that the authors identify and support such a choice.*

R3: In the revised manuscript, we have specified that primers 515F and 907R amplify

the V4–V5 region of the 16S rRNA gene, which offers broad coverage and reliable taxonomic resolution for bacterial community profiling (Lines 158–160) based on previous study

Revised in text (Lines 163–165): “After electrophoresis and concentration analysis, the qualified DNA samples were sent to Guangdong Magigene Biotechnology Co., Ltd. (Guangzhou, China) for high-throughput 16S rRNA sequencing using primers 515F (5′-GTGCCAGCMGCCGCGGTAA-3′) and 907R (5′-CCGTCAATTCMTTTRAGTTT-3′). These primers amplify the V4–V5 region of the 16S rRNA gene, which provides broad coverage and reliable taxonomic resolution for bacterial community profiling²³.”

Q4: In the "Results and Discussion: The succession of the bacterial community drives heat production during HTC", the authors wrote that the temperature reached a peak of 94dC within 24 h. However-, in the Figure, it took approximately 2.5 days to peak at 94dC. Please clarify.

R4: We sincerely apologize for the incorrect expression in the result. Although HTC composting can reach extremely high temperatures within 24 hours, the rate of temperature increase depends on factors such as the composting materials and the ambient temperature. We have corrected this error to reflect the actual data shown in the figure.

Revised in text (Lines 247–248): “The temperature increased rapidly, reaching a peak of 94°C within 3 days, and remained above 80°C for 9 days during the hyperthermophilic phase (D4).”

References

6. Liao H, Lu X, Rensing C, et al. Hyperthermophilic composting accelerates the removal of antibiotic resistance genes and mobile genetic elements in sewage sludge. *Environ Sci Technol* 52, 266-276 (2018).
8. Yamada T, Miyauchi K, Ueda H, et al. Composting cattle dung wastes by using a hyperthermophilic pre-treatment process: Characterization by physicochemical and molecular biological analysis. *J Biosci Bioeng* 104, 408-415 (2007).
9. Oshima T, Moriya T. A preliminary analysis of microbial and biochemical properties of high-temperature compost. *Ann N Y Acad Sci* 1125, 338-344 (2008).
12. Yu Z, Tang J, Liao H, et al. The distinctive microbial community improves composting efficiency in a full-scale hyperthermophilic composting plant. *Bioresour Technol* 265, 146-154 (2018).
21. Liao H, Lu X, Rensing C, et al. Hyperthermophilic composting accelerates the removal of antibiotic resistance genes and mobile genetic elements in sewage sludge. *Environ Sci Technol* 52, 266-276 (2018).
22. Tortosa G, Albuquerque JA, Ait-Baddi G, et al. The production of commercial organic amendments and fertilisers by composting of two-phase olive mill waste (“alperujo”). *J Cleaner Prod* 26, 48-55 (2012).
23. Claesson MJ, Wang Q, O'Sullivan O, et al. Comparison of two next-generation sequencing technologies for resolving highly complex microbiota composition using tandem variable 16s rRNA gene regions. *Nucleic Acids Res* 38, e200-e200 (2010).
35. Finore I, Feola A, Russo L, et al. Thermophilic bacteria and their thermostable enzymes in composting processes: A review. *Chem Biol Technol Agric* 10, 7 (2023).
39. Wang Z, Chen Z, Jiang S, et al. Improvement of metabolic heat accumulation for hyperthermophilic composting system: Influencing factors and microbial communities. *Environ Sci Eur* 37, 74 (2025).

Re: mSystems00956-25R1 (**Metabolic activity and survival strategies of thermophilic microbiomes during hyperthermophilic composting**)

Dear Prof. Hanpeng Liao:

Revision Guidelines

Sincerely,
Xue Guo
Editor
mSystems

Reviewer #1 (Comments for the Author):

The manuscript has improved significantly through these revisions. Below are minor suggestions.

Introduction:

Lines 105-112: The expansion with specific examples (e.g., sporulation, shifts in cellular behavior, stress response pathways) is valuable. However, the subsequent description of the current study's results does not logically form the basis for introducing this

hypothesis. Consider removing these result details here. Instead, briefly describe relevant research strategies used in prior studies of other systems (Lines 88-99) to provide better context for proposing this hypothesis.

Materials and Methods:

Lines 144-147: Abbreviations for the physicochemical properties are introduced in Lines 141-144. For consistency and clarity, please use these defined abbreviations exclusively when referring to these properties throughout the rest of the Methods section

Results and Discussion:

Lines 389-390: The statement suggesting "a reduced representation of nutrient cycling function" appears less conclusive based solely on the data presented in Fig. S9. Please re-evaluate this specific conclusion in light of Fig. S9's evidence or provide additional justification linking the figure data explicitly to this functional claim.

Reviewer #2 (Comments for the Author):

I thank the authors for clarifying point by point. I only have a minor comment: In Figure S2, the nodes for bacterial community are confusing. Do they signify all genera with relative abundance of $>0.1\%$? I think the figure should be improved to be more straightforward.

Responses to the reviewer's comments

Editors' comments to the author:

Reply: We sincerely thank you for the opportunity to revise our manuscript and for forwarding valuable feedback from the reviewers. We have carefully considered all the comments and have revised the manuscript accordingly. A detailed, point-by-point response to each comment is provided below. Changes made in the manuscript are highlighted in yellow for clarity. We hope the revised version addresses all the concerns and meets the journal's standards for publication.

Reviewer #1 (Comments to the Author):

The manuscript has improved significantly through these revisions. Below are minor suggestions.

R (Reply): We sincerely thank the reviewer for the positive evaluation of our revised manuscript and for acknowledging the improvements made. We have carefully addressed each point as detailed below.

Major comments:

Q1 (Question 1): *Lines 105-112: The expansion with specific examples (e.g., sporulation, shifts in cellular behavior, stress response pathways) is valuable. However, the subsequent description of the current study's results does not logically form the basis for introducing this hypothesis. Consider removing these result details here. Instead, briefly describe relevant research strategies used in prior studies of other systems (Lines 88-99) to provide better context for proposing this hypothesis.*

RI: We fully agree that including our study results in the section proposing the hypothesis caused unnecessary confusion between background and findings. In the revised manuscript, we removed these result descriptions (Lines 105–110) and retained only general examples of thermophilic survival strategies. We also clarified

the preceding paragraph (Lines 92–97) by summarizing how genome-resolved metagenomics and metatranscriptomics have been used in geothermal environments to reveal key survival mechanisms such as sporulation, stress responses, and shifts in cellular behavior.

Revised in text (lines 92–97): “Recent advances in genome-resolved metagenomics and metatranscriptomics have enabled researchers to reconstruct microbial genomes and characterize transcriptional activity in extreme environments such as hot springs and hydrothermal vents¹⁵⁻¹⁸. These studies have uncovered a wide array of survival strategies employed by thermophilic microorganisms, including sporulation, stress response activation, and shifts in cellular behavior, which reflect long-term evolutionary adaptation to persistent thermal stress^{17, 19, 20},”

Revised in text (lines 105–110): “We hypothesized that: (1) core thermophilic taxa act as primary drivers of heat production; and (2) thermophilic bacteria employ a range of physiological and molecular strategies to cope with extreme thermal stress during HTC, such as sporulation, shifts in cellular behavior, and activation of stress response pathways. This study provides genome-resolved insights into the metabolic heat generation and survival strategies of key thermophiles in HTC, with implications for optimizing microbial processes in high-temperature organic waste treatment systems.”

Q2: Lines 144-147: Abbreviations for the physicochemical properties are introduced in Lines 141-144. For consistency and clarity, please use these defined abbreviations exclusively when referring to these properties throughout the rest of the Methods section.

R2: We have revised the manuscript to ensure consistent use of the defined abbreviations for physicochemical properties throughout the text.

Revised in text (lines 39–140): “WC was determined by drying compost samples at 105 °C for 24 hours using the drying oven. NH₄⁺ and NO₃⁻ were extracted using 2 M KCl and measured with a continuous-flow autoanalyzer after a 10-fold dilution

(FlowSys, Systea, Rome, Italy).”

Q3: Lines 389-390: The statement suggesting "a reduced representation of nutrient cycling function" appears less conclusive based solely on the data presented in Fig. S9. Please re-evaluate this specific conclusion in light of Fig. S9's evidence or provide additional justification linking the figure data explicitly to this functional claim.

R3: We acknowledge that the original conclusion was premature, as it relied solely on gene content comparison without integrating transcriptional activity data. In the revised manuscript, we have rephrased this section to better reflect the available evidence. Specifically, we now emphasize that although thermophilic MAGs encode fewer nutrient cycling genes than mesophilic MAGs (Fig. S9), dominant thermophilic populations such as *Thermus* and *Planifilum* show significantly elevated transcriptional activity during the high-temperature phase (Fig. S7). This contrast may reflect a survival strategy in which these populations prioritize thermal adaptation over nutrient metabolism to maintain ecological competitiveness under extreme heat stress.

Revised in text (Lines 384–388): “Genomic analysis revealed that mesophilic MAGs harbor significantly more genes related to carbon and nitrogen cycling than thermophilic MAGs ($p < 0.01$, Fig. S9), suggesting that mesophiles likely have greater potential to drive carbon and nitrogen turnover during the HTC composting process compared to thermophiles. In contrast, dominant thermophilic lineages such as *Thermus* and *Planifilum* exhibited markedly elevated transcriptional activity during the high-temperature phase (Fig. S8). This pattern may reflect a survival strategy in which these thermophilic populations prioritize thermal adaptation over nutrient metabolism to maintain ecological competitiveness. Consistent with this interpretation, previous findings suggest that thermophiles may have streamlined their genomes—losing certain metabolic pathways—to better adapt to high-temperature environments⁴⁶”

Reviewer #2 (Comments for the Author):

I thank the authors for clarifying point by point. I only have a minor comment: In Figure S2, the nodes for bacterial community are confusing. Do they signify all genera with relative abundance of >0.1%? I think the figure should be improved to be more straightforward:

R: We appreciate the reviewer's comment and agree that the original network visualization may have been unclear. To improve clarity, we revised Figure S2 by replacing the red node labels with the corresponding physicochemical variable names. Additionally, we updated the figure legend to clarify that the Mantel test was based on a Bray–Curtis dissimilarity matrix of bacterial community composition (constructed from all genera with >0.1% relative abundance) and a Euclidean distance matrix of physicochemical properties.

Revised in figure S2:

Figure S2. Correlations between physicochemical properties and bacterial community composition based on 16S rRNA gene amplicon sequencing. For the Mantel test, bacterial community composition was represented by a Bray–Curtis dissimilarity matrix calculated from genus-level relative abundances (>0.1%), and physicochemical properties were represented by Euclidean distance matrices. Edge width represents the absolute value of the Mantel correlation coefficient, and edge color indicates statistical significance. gradients denoting Pearson's correlation coefficients. Abbreviations: ammonium (NH₄⁺); Temperature (Temp) The upper triangular matrix shows pairwise correlations among physicochemical properties, with color; total

organic carbon (TOC); dissolved total nitrogen (DN); electrical conductivity (EC); inorganic carbon (IC); nitrate (NO_3^-); water content (WC); total carbon (TC), total nitrogen (TN).

References

15. Dong X, Zhang C, Peng Y, *et al.* Phylogenetically and catabolically diverse diazotrophs reside in deep-sea cold seep sediments. *Nat Commun* **13**, 4885 (2022).
16. Wu X, Cui Z, Peng J, *et al.* Genome-resolved metagenomics identifies the particular genetic traits of phosphate-solubilizing bacteria in agricultural soil. *ISME Commun* **2**, 17 (2022).
17. Zhang Y, Liu T, Li M-M, *et al.* Hot spring distribution and survival mechanisms of thermophilic comammox nitrospira. *The ISME Journal* **17**, 993-1003 (2023).
18. Hauptfeld E, Pappas N, van Iwaarden S, *et al.* Integrating taxonomic signals from mags and contigs improves read annotation and taxonomic profiling of metagenomes. *Nat Commun* **15**, 3373 (2024).
19. Ward L, Taylor MW, Power JF, *et al.* Microbial community dynamics in inferno crater lake, a thermally fluctuating geothermal spring. *The ISME journal* **11**, 1158-1167 (2017).
20. Chen X, Tang K, Zhang M, *et al.* Genome-centric insight into metabolically active microbial population in shallow-sea hydrothermal vents. *Microbiome* **10**, 170 (2022).
46. O'Donnell DR, Hamman CR, Johnson EC, *et al.* Rapid thermal adaptation in a marine diatom reveals constraints and trade-offs. *Global Change Biol* **24**, 4554-4565 (2018).

Re: mSystems00956-25R2 (**Metabolic activity and survival strategies of thermophilic microbiomes during hyperthermophilic composting**)

Dear Prof. Hanpeng Liao:

Your manuscript has been accepted, and I am forwarding it to the ASM production staff for publication. Your paper will first be checked to make sure all elements meet the technical requirements. ASM staff will contact you if anything needs to be revised before copyediting and production can begin. Otherwise, you will be notified when your proofs are ready to be viewed.

Sincerely,
Xue Guo
Editor
mSystems